# Functional Study of *Haemophilus ducreyi* Cytolethal Distending Toxin Subunit B

**DOI:** 10.3390/toxins12090530

**Published:** 2020-08-19

**Authors:** Benoît J. Pons, Nicolas Loiseau, Saleha Hashim, Soraya Tadrist, Gladys Mirey, Julien Vignard

**Affiliations:** Toxalim (Research Center in Food Toxicology), Université de Toulouse, INRAE, ENVT, INP-Purpan, UPS, 31300 Toulouse, France; B.Pons@exeter.ac.uk (B.J.P.); nicolas.loiseau@inrae.fr (N.L.); saleha.hashim@inrae.fr (S.H.); souria.tadrist@inrae.fr (S.T.)

**Keywords:** Cytolethal Distending Toxin, CdtB, phosphatase, nuclease, DNA damage, cytotoxicity, cell cycle checkpoint, site-directed mutagenesis

## Abstract

The Cytolethal Distending Toxin (CDT) is produced by many Gram-negative pathogenic bacteria responsible for major foodborne diseases worldwide. CDT induces DNA damage and cell cycle arrest in host-cells, eventually leading to senescence or apoptosis. According to structural and sequence comparison, the catalytic subunit CdtB is suggested to possess both nuclease and phosphatase activities, carried by a single catalytic site. However, the impact of each activity on cell-host toxicity is yet to be characterized. Here, we analyze the consequences of cell exposure to different CDT mutated on key CdtB residues, focusing on cell viability, cell cycle defects, and DNA damage induction. A first class of mutant, devoid of any activity, targets putative catalytic (H160A), metal binding (D273R), and DNA binding residues (R117A-R144A-N201A). The second class of mutants (A163R, F156-T158, and the newly identified G114T), which gathers mutations on residues potentially involved in lipid substrate binding, has only partially lost its toxic effects. However, their defects are alleviated when CdtB is artificially introduced inside cells, except for the F156-T158 double mutant that is defective in nuclear addressing. Therefore, our data reveal that CDT toxicity is mainly correlated to CdtB nuclease activity, whereas phosphatase activity may probably be involved in CdtB intracellular trafficking.

## 1. Introduction

The Cytolethal Distending Toxin (CDT) is a genotoxic virulence factor found in several Gram-negative pathogenic bacteria, such as *Escherichia coli* (*E. coli*) [1], *Campylobacter* spp. [2], *Aggregatibacter actinomycetemcomitans* (*A. actinomycetemcomitans*) [3,4], and *Haemophilus ducreyi* (*H. ducreyi*) [5]. CDT-producing bacteria are associated with many inflammatory diseases such as chancroid, enterocolitis, and periodontitis (reviewed in [6]). In such contexts, CDT has been shown to favor bacterial long-term persistence and inflammation [7,8,9,10,11,12], and to promote colorectal carcinogenesis [13,14,15] in mice models. CDT has been named according to the phenotype of exposed cells: Cellular and nuclear distention [1], preceding apoptotic cell death [16,17]. CDT cytotoxicity has been shown to rely on DNA damage induction [18], more precisely single- and double-strand breaks [19], resulting in the activation of the DNA damage response [20,21] and, therefore, cell cycle arrest [22,23].

CDT is a tripartite toxin presenting an AB_2_ structure [24,25,26]: The CdtA and CdtC (B moieties) subunits are responsible for the delivery of the catalytic CdtB (A moiety) subunit in host cells, which in turn employs retrograde trafficking across different subcellular compartment to reach the nucleus [27,28]. Of note, the typhoid toxin produced by *Salmonella enterica* serovar Typhi presents an A_2_B_5_ structure: Five regulatory PltB subunits and two catalytic subunits, PltA and CdtB [29]. CdtB, being the catalytic subunit of both the CDT and typhoid toxins, carries the toxin biochemical activities and is, therefore, responsible for the cellular toxin effects. Indeed, direct delivery of the CdtB subunit in cultured cell reproduces the holotoxin-induced cell cycle arrest [30,31,32,33], DNA damage [34], or apoptosis [21,31,35,36,37,38,39]. Moreover, point mutations on CdtB catalytic residues annihilate the toxic effects of the holotoxin in cultured cells [24,36,40]. However, the exact biochemical activity of CdtB is not fully characterized, and, therefore, the detailed mode of action of CDT during infection and pathogenesis is still unclear.

The different CdtB sequences exhibit homology with proteins belonging to the Mg^2+^/Mn^2+^-dependent endonuclease-exonuclease-phosphatase (EEP) family, such as bovine DNase I [36,40,41]. CdtB and DNase I also exhibit an overall 3D structure homology and good conservation of several key residues involved in catalytic activity and metal or substrate binding [24,42]. Due to these sequence and structure similarities, CdtBs were speculated to present a nuclease activity [24,40]. To assess this potential activity, an in vitro plasmid digestion assay, initially developed for DNase I [43], has been proposed [40] and widely used [21,24,30,37,44,45,46,47,48,49]. Nonetheless, there are important discrepancies between in vitro observed activity and cellular effects. Indeed, catalytic residues mutations completely abolished CDT- and CdtB-mediated DNA damage in host cells but did not hamper CdtB plasmid digestion [34]. Moreover, the non-catalytic CdtA and CdtC subunits, which are unable to induce DNA damage on their own, exhibit a plasmid degradation activity similar to CdtB [34,50]. Taken together, these results challenge the relevance of this in vitro assay to study CdtB nuclease activity.

On the other hand, sequence similarities were found between *A. actinomycetemcomitans* or *Helicobacter hepaticus* (*H. hepaticus*) CdtBs (AactCdtB and HhepCdtB, respectively) and the inositol polyphosphate 5-phosphatase domain (IP5P) of *Schizosaccharomyces pombe* (*S. pombe*) synaptojanin, another EEP family member [44,48]. Therefore, it has been hypothesized that AactCdtB exhibits phosphatase activity, and biochemical studies indicated that AactCdtB can dephosphorylate phosphatidylinositol-3,4,5-trisphosphate (PIP_3_) in position 5 to produce phosphatidylinositol-3,4-bisphosphate (PI3,4P_2_) [48]. Since catalytic and metal or substrate binding residues are highly conserved between CdtB, DNase I, and IP5P, this suggests that the two putative CdtB activities, nuclease and phosphatase, share the same catalytic site. Which of these two activities is essential for CDT-related pathogenicity is still under debate. Indeed, depending on the host cell type and the CDT bacterial origin, each of these biochemical CdtB activities have been more directly implicated in cellular toxicity [49,51]. However, their exact function and relationship during the course of cell intoxination still need to be clarified.

Here, we decipher the importance of several CdtB key residues during CDT intoxination in human cells. Besides targeting catalytic or metal-binding residues, several *H. ducreyi* CdtB (HducCdtB) mutants were generated in an attempt to uncouple nuclease and phosphatase activities, mainly by targeting residues potentially involved in substrate binding. Based on cellular sensitivity to holotoxin, two classes of mutants emerged. Class I is composed of CDTs devoid of any activity, bearing mutations in catalytic (H160A) and metal binding (D273R) residues, or mutations in residues previously reported to be involved in DNA binding (R117A, R144A, and N201A triple mutant referred to as RRN). Conversely, mutation of residues theoretically involved in phosphatase activity (F156, T158, and A163) only partly abolished cellular defects, defining the mutant class II. Moreover, we identified the G114 residue as potentially important for lipid substrate binding, whose corresponding CDT mutant also belongs to class II. As we postulate phosphatase activity could be involved in lipid modification and, therefore, in CdtB retrograde transport, this normal trafficking route was circumvented by direct CdtB delivery through transfection [34]. Interestingly, CdtB transfection restores the genotoxic activity of the class II mutants, with the exception of CdtB mutated on F156 and T158 (named FT). Indeed, we bring evidence that FT mutant is defective in nuclear localization, and demonstrate that forcing its translocation to the nucleus enables DNA damage induction. In conclusion, our results indicate that CDT toxicity is strongly linked to DNA damage induction, mediated by CdtB nuclease activity, but this depends on the proper CdtB transport to the nucleus, in which phosphatase activity is probably involved.

## 2. Results

### 2.1. Mutation Choice

To gain insight into the function of residues potentially involved in CdtB catalytic activities, several HducCdtB mutants were generated on the basis of previous reports (Table 1) and structural modelization. The HducCdtB structure [24] (PBD ID: 1SR4) was aligned both with human Type II Inositol 1,4,5-trisphosphate 5-Phosphatase (INPP5B) co-crystalized with PI3,4P_2_ [52] and with bovine DNase I co-crystallized with a DNA octamer [53] (Appendix A), in order to model the interaction of HducCdtB with these two putative substrates and to locate and characterize the amino-acid residues of interest analyzed in this study (Figure 1).

First, two residues crucial for CdtB activity were targeted. Indeed, the catalytic histidine H160 [24] was mutated to alanine (H160A), as analogous substitutions were shown to inactivate CdtBs from diverse bacterial origins [40,55]. The D273 residue, involved in metal binding, was also mutated to arginine (D273R) to abolish HducCdtB activity as previously published [54].

Then, various substrate-binding mutants were engineered in order to uncouple nuclease and phosphatase activities. Three amino acids, R117, R144, and N201, originally described as DNA binding residues [24], were substituted with three alanine (R117A-R144A-N201A, noted RRN). On the other hand, docking analyses suggested that AactCdtB A163 interacts differently with inositol-3,4,5-trisphosphaste and inositol-3,4-bisphosphaste [49]. Moreover, structural alignment revealed that HducCdtB A163 corresponded to INPP5B A403 that recognized the inositol ring of PI3,4P_2_ [52], strengthening the hypothesis of a role in phosphatidylinositol binding. The same study also identified AactCdtB F156 and T158 as potentially involved in phosphatase activity on the basis of a sequence comparison between AactCdtB, DNase I, and IP5P, despite a certain distance from the catalytic pocket (Figure 1). Therefore, the A163R and F156I-T158I (FT) mutants were generated. Finally, the modelized interaction of HducCdtB with DNA or PI3,4P_2_ drew our attention towards G114, located on a loop potentially involved in binding both substrates. The corresponding hydrophobic loop in INPP5B contained the lipid chain 1 recognition motif (LC1R) that interacts with the aliphatic region of the lipid substrate [52], suggesting a similar function in CdtB. As the distance between the non-polar HducCdtB G114 and the hydrocarbon chain of PI3,4P_2_ should be around 2 angstroms (Figure 1b), compatible with a possible interaction, this residue has thus been substituted with polar threonine (G114T) to destabilize the putative interaction with PI3,4P_2_ (Figure 1b).

### 2.2. Cytotoxic Activity of the CdtB Mutants in the Holotoxin Context

The purified WT and mutants CdtBs (Appendix A) were incubated with CdtA and CdtC subunits to reassemble active holotoxins [34]. The cytotoxic activity of these recombinant CDTs was assessed in HeLa cells by clonogenic assay after 10 days of treatment (Figure 2a). While the WT toxin induces a dose-dependent viability loss, with more than 99.9% cell death at 5 ng/mL, two classes of mutants emerged. The class I, composed by H160A, D273R, and RRN, is devoid of any cytotoxic activity at the highest toxin concentration. Besides, every single mutation from the RRN background confers the same defects as the triple mutant in the clonogenic assay, but also in all the other assays performed in this study (Appendix A). On the other hand, mutants potentially impaired in phosphatase activity, namely G114T, FT, and A163R, only displayed reduced cytotoxic activity compared to WT, defining the class II mutants. Then, cell cycle analyses were conducted after 36 h of exposure to 35 ng/mL of CDT. When incubated with the WT toxin, the vast majority of HeLa cells were blocked at G2/M (Figure 2b). Conversely, cells exposed to class I mutants exhibit the same cell cycle profile compared to untreated cells, confirming that these mutations abolished CdtB activity. Treatment with mutants from class II attenuated CDT-induced G2/M arrest for G114T and A163R, whereas FT did not cause obvious cell cycle defects under these conditions. At a lower toxin concentration of 3.5 ng/mL, only WT CDT was still able to impede cell cycle progression, at a level close to A163R at 35 ng/mL, indicating that class II mutants were at least 10-fold less active than WT in HeLa cells. Thus, our data suggest that similar to H160 catalytic residue, targeting residues involved in metal or DNA binding completely abrogates CdtB cytotoxic activity. Alternatively, mutation of residues possibly implicated in phosphatase activity partly alters CDT-mediated cellular defects.

Lymphoblastoid cell lines have been shown to be extremely sensitive to CDT [6], in a way that may depend on CdtB phosphatase activity in Jurkat cells [49]. To test this hypothesis, we first attempted to evaluate the phosphatase activity of the CdtB mutants through an indirect cellular assay focusing on the PIP_3_/AKT signaling pathway deregulation. Indeed, it has been previously reported that AactCDT-induced PIP_3_ dephosphorylation decreases the pool of intracellular phosphorylated AKT (p-AKT) [49]. However, no obvious alteration of p-AKT level could be observed in the Jurkat cells treated for 2 or 4 h with various toxin concentrations (Appendix A), or by comparing WT CDT with class I or class II mutants (Appendix A). Thus, we were not able to evaluate any perturbation of the CdtB phosphatase activity. Then, a dose-dependent analysis of CDT-mediated cell cycle arrest was conducted in the Jurkat cells (Figure 3). Our results indicated that 1 ng/mL of WT CDT induced a similar G2/M block than 35 ng/mL of WT CDT in HeLa cells, confirming the highest sensitivity of Jurkat cells. As expected, class I mutants have lost their ability to block the Jurkat cell cycle, as shown in HeLa. On the other hand, class II mutations did not totally abolish CDT-induced G2/M arrest. Indeed, A163R and G114T mutants completely blocked the Jurkat cell cycle at 35 ng/mL, but only partly arrested cells at G2/M at 1 ng/mL, at a lower level compared to 0.35 ng/mL of WT. This supports that class II mutations decrease CDT activity in Jurkat cells around 10 times, as previously observed in HeLa cells. Cell cycle perturbation with the FT mutant was only apparent at the highest tested concentration of 35 ng/mL, to an extent similar to 1 ng/mL of G114T but inferior to 0.35 ng/mL of WT. Taken together, these results demonstrated that while Jurkat cells were more sensitive to CDT than HeLa cells, class I mutations, targeting catalytic, metal binding, and DNA binding residues, completely abrogated CDT activity in both cell lines. Alternatively, class II mutations, potentially affecting CdtB phosphatase activity, decreased CDT-mediated cell cycle defects in both Jurkat and HeLa cells by about 90%.

### 2.3. Effects of the CdtB Mutations on CDT-Induced DNA Damage

To further compare the defects associated to CdtB mutations, we evaluated their impact on DNA damage induction by monitoring γH2AX increase—a well-established DNA damage biomarker [56]—through immunofluorescence analysis, in HeLa cells treated for 24 h with 35 ng/mL of CDT (Figure 4 and Appendix A). The γH2AX signal was quantified individually in each cell, as previously described [34]. The WT CDT globally induced a strong 16,8-fold increase of the γH2AX signal compared to control cells, with almost all cells being damaged. In accordance with cytotoxicity analysis, class I mutants (H160A, D273R, and RRN) did not show any genotoxic effect. For class II mutants, A164R and G114T equally reduced the CDT-mediated γH2AX signal to approximately 40% of the WT. This implies that fewer cells accumulated detectable DNA damage, representing around 75% of the entire cell population. In the same way as the clonogenicity assay and cell cycle analysis, the toxin bearing the FT mutation was unable to increase the γH2AX signal under these conditions. Overall, class I mutations targeting catalytic, metal binding, or DNA binding residues completely inhibited the genotoxic and cytotoxic activities of HducCDT while A163R and G114T class II mutants only reduced them. Conversely, the FT mutant was unable, or more probably strongly impaired, in its capacity to induce DNA damage or cell cycle arrest while still retaining some cytotoxic activity. Taken together, these results indicate that HducCDT cytotoxicity correlates with CdtB nuclease activity. However, this also suggests that CDT-mediated DNA damage at least partly depends on residues potentially involved in CdtB phosphatase activity.

### 2.4. Direct CdtB Delivery

We speculated that CdtB transport from the host cell membrane to the nucleus may involve its phosphatase activity. In order to bypass the intracellular trafficking pathway that normally arises during CDT intoxination, CdtB has been directly delivered inside host cells in the absence of CdtA and CdtC [34]. HeLa cells were transfected with 1 µg/mL of recombinant CdtB for 36 h before cell cycle analysis (Figure 5a). Transfection of WT CdtB recapitulated the cell cycle defects observed when exposing cells to 3.5 ng/mL of WT holotoxin (Figure 2). Class I and FT mutants did not affect the cell cycle, whereas cells transfected with A163R exhibited a slight G2/M block. Interestingly, the cell cycle profile of cells transfected with G114T was the same as WT CdtB, demonstrating that this artificial CdtB delivery system restored the cytotoxic activity of G114T. The same strategy was then used for γH2AX immunofluorescence analyses (Figure 5b and Appendix A). After 24 h of transfection with 1 µg/mL of WT CdtB, HeLa cells exhibited an approximately 5-fold increase of γH2AX global signal or γH2AX-positive cells, compared to control or to cells transfected with class I or FT mutants. Interestingly, the A163R and G114T mutants were not significantly different from the WT. These data indicate that G114T, and to a lesser extent A163R, retrieved their capacity to induce DNA damage and cell cycle arrest when artificially introduced in host cells, in contrast to the holotoxin context. As these mutants are potentially defective in phosphatase activity, this supports that this activity could be important for CdtB trafficking. On the other hand, the FT mutant presents the very same defects, whether transfected alone or associated with CdtA and CdtC, suggesting it may be affected by other functions compared to other class II mutants.

### 2.5. CdtB Cellular Expression

In order to determine whether CdtB defects associated with FT mutations could result from impaired subcellular localization, a direct expression strategy in host cells was conducted, ensuring an important amount of intracellular CdtB and massive but non-quantifiable DNA damage induction [34]. CdtB cDNAs were subcloned in mammalian expression plasmid in C-terminal fusion with mCherry before transfection in HeLa cells for 14 h. As a control for CdtB mislocalization, the previously characterized nuclear localization signal (NLS), corresponding to residues 114–124 [38], had been deleted (∆NLS mutant). According to the mCherry signal, the ∆NLS mutant exhibited a different sublocalization pattern compared to WT (Figure 6 and Appendix A). Indeed, while WT CdtB was normally enriched in the nucleus, the absence of NLS disturbed this nuclear import. Moreover, the intense γH2AX staining observed after WT CdtB expression was lost with ∆NLS. Class I mutants showed a strong nuclear enrichment but did not exert any genotoxic activity. Conversely, A163R and G114T mutants displayed normal subcellular localization and increased γH2AX signal similarly to WT, confirming that forcing CdtB entry in host cells through artificial routes alleviates the defects induced by A163 or G114 mutations. Interestingly, the nuclear localization of the FT mutant was impaired and close to ∆NLS. Moreover, DNA damage induction was also lower, as revealed by the clear γH2AX signal decrease, suggesting that the defects induced by these mutations could be attributable to CdtB mislocalization. To test this hypothesis, CdtB-mCherry was fused in the N-terminal position to the chromatibody (Cb), a camelid single-domain antibody specifically directed against the histone H2A/H2B heterodimer, ensuring nuclear enrichment and chromatin binding [57]. As shown with the ∆NLS mutant, adding the chromatibody relocated the protein to the nucleus, although without retrieving DNA damage induction. Chromatibody fusion to WT further increased CdtB nuclear fraction, without obviously affecting γH2AX staining. However, chromatibody fusion to FT greatly enhanced the nuclear localization to the chromatibody-WT level and restored γH2AX induction. Therefore, these data demonstrate that the FT mutation affects CdtB nuclear addressing, impeding the subsequent induction of genomic DNA damage.

## 3. Discussion

CDT is involved in the virulence properties of many pathogenic bacteria by promoting invasion, inflammation, and persistent colonization [7,8,9,10,58]. Consequently, CDT directly impacts different pathologic outcomes, such as ulcer development in *H. ducreyi*-induced chancroid [33], aggressive periodontal disease associated with *A. actinomycetemcomitans* [59], or even carcinogenesis [13,15]. However, the biochemical mechanisms underlying CDT-induced pathologies are not fully characterized. The catalytic subunit, CdtB, presents sequence and structural homology with bovine DNase I, thus suggesting a nuclease activity [24,36,40]. Moreover, sequence alignment with IP5P domain of *S. pombe* synaptojanin led to propose that CdtB subunits may possess both nuclease and phosphatase activities, carried by a single catalytic site [48,49]. The precise role of each CdtB putative activity during CDT intoxination still needs to be elucidated. However, the biochemical characterization of CdtB nuclease activity mostly relies on plasmid digestion assays [24,40,49] that we recently showed to be inappropriate for this protein [34]. Defining novel strategies to evaluate CdtB activities is, therefore, of prime importance. Here, we propose to use the same cell-based approaches that we previously described [34] to estimate the impact of CdtB mutations on host cell responses and relate them to CdtB catalytic activities.

To assess nuclease activity, the cellular assay performed here consists of the direct CdtB delivery into the host cell, bypassing the prerequisite of CDT binding subunits and retrograde transport [34]. This test, allowing for a quantitative comparison of different CdtBs, has been used here to analyze mutants potentially affected in one or both catalytic activities. Our results confirm the loss of activity for mutants targeting catalytic (H160) or metal binding (D273) residues but challenge other reports that classified some CdtB mutants on the basis of plasmid digestion assays. For instance, the *H. ducreyi* RRN triple mutant that initially presented as defective in DNA binding and thus nuclease deficient [24], was then shown to be more active compared to WT for A. *actinomycetemcomitans*, with every single mutant displaying distinctive phenotypes, either hyperactive, inactive, or unaffected [49]. Our data indicate that none of the RRN single or triple mutants induce DNA damage in living cells, implying that all three DNA binding residues are essential for CdtB nuclease activity. Besides, we cannot rule out the possibility that they also play a role in lipid substrate binding, as previously suggested [49]. Then, A163, originally predicted to be involved in lipid substrate binding, was ultimately found to be absolutely essential in plasmid digestion assay, and, therefore, identified as required for nuclease activity [49]. However, we show here that the A163R mutant induces almost the same γH2AX signal than WT when artificially introduced in cells, demonstrating that this mutant is still able to induce DNA damage. Besides, the attenuated cell cycle defects observed with the A163R mutant compared to the WT or G114T mutant might result from the slight DNA damage reduction. Finally, this approach allowed us to establish that the G114 mutation does not affect CdtB genotoxic potential. Therefore, we present here evidence that the defects associated with A163R or G114T mutations cannot be attributed to impaired nuclease activity.

The defects associated with FT mutations are atypical. The F156 and T158 residues of CdtB, conserved with IP5P, were, respectively, substituted to I and A residues to match the DNase I sequence in an effort to hamper phosphatase activity specifically [49]. Based on the plasmid digestion assay, this mutant was shown to exhibit higher nuclease activity while defective in phosphatase activity, and to be inactive in cells. Unexpectedly, we demonstrate here that contrary to the A163R and G114T putative phosphatase mutants, FT direct delivery into host cells does not induce a detectable level of γH2AX signal. Of note, only a slight γH2AX increase was detected after transient FT expression driven by the strong CMV enhancer (Figure 6). This strategy ensures substantial DNA damage induction compared to holotoxin treatment or CdtB direct delivery, at least at the concentrations tested here. Hence, FT genotoxic activity is not totally abrogated as a class I mutant but rather more affected than A163R and G114T. FT-related defects must, therefore, be distinguished from the other class II mutants. Indeed, FT inability to increase γH2AX relies on impaired nuclear localization, as DNA damage induction can be restored by fusing CdtB to the chromatibody, a small camelid antibody fragment promoting chromatin binding, and thus nuclear retention of proteins that may passively diffuse through the nucleus, and potentially nuclear addressing [57]. In contrast, chromatibody correctly targets ∆NLS mutant to the nucleus but does not rescue DNA damage induction, indicating that this mutant is also nuclease-defective. Indeed, the deleted NLS sequence includes the R117 residue that we show here to be essential for CdtB nuclease activity. Besides, the apparent discrepancy between (1) the similar cytotoxic activity between FT and the other class II mutants observed in the clonogenic assay, and (2) the lower FT-induced DNA damage and cell cycle defects, might rely on distinct timing of exposure, which is 10 days for clonogenic survival and from 12 to 36 h for the other assays. This suggests that FT nuclear addressing could be rather delayed than strictly impaired. To conclude, our data demonstrate that F156 and T158 residues are necessary for proper CdtB localization to the nucleus, which is a prerequisite to CDT-related genotoxicity.

One of the main findings of this study concerns the differences observed between reconstituted holotoxins and artificially internalized CdtBs bearing A163R or G114T mutations. Indeed, while these CDT mutants clearly induce less cell death, cell cycle arrest, and DNA damage, CdtB transfection restores cell cycle defects and DNA damage, especially for G114T. These differences foster several assumptions. First, our data demonstrate that restoring CdtB genotoxic potential through direct cell delivery also rescues G2 arrest, emphasizing a good correlation between DNA damage and cell cycle defects. Hence, the CDT-induced cell cycle block directly results from its genotoxic activity. Then, given that CdtBs carrying the A163R or G114T mutations retain nuclease activity comparable to WT, they must indirectly impede DNA damage induction in the holotoxin context. The rationale behind targeting these residues is based on their putative role in lipid substrate binding. We speculate that A163R and G114T mutants are defective in phosphatase activity. In light of this hypothesis, we infer that phosphatase activity is essential for DNA damage induction during normal CDT intoxination by promoting CdtB internalization. Indeed, CdtB from *A. actinomycetemcomitans* has been shown to dephosphorylate PIP_3_ to produce PI3,4P_2_ [48,49], an important factor controlling endosomal trafficking [60]. In an attempt to evaluate CdtB phosphatase activity, we favored an indirect cellular approach because biochemical testing brought contradictory results [48,50], with the discrepancies related to CdtB nuclease activity analysis [34]. The CDT-induced perturbation of the PIP_3_/PI3,4P_2_ pool has been shown to decrease phosphorylation of the downstream target AKT in Jurkat cells [49]. Unfortunately, we were unable to reproduce these results, and no reliable alteration of the p-AKT level was observed in the Jurkat cells treated with CDT. However, it has to be noticed that the p-AKT level increase has been reported upon *H. ducreyi* CDT treatment of HCT116 cells [61] or *H. hepaticus* CdtB expression in engrafted mice tumors [62]. This AKT phosphorylation might be a response to CdtB-induced DNA damage. Indeed, phosphoinositides have been shown to accumulate at DNA damage sites [63], and AKT is phosphorylated in response to DNA damaging agents such as etoposide or bleomycin [64] in a manner depending on Ataxia Telangiectasia Mutated (ATM)—a major kinase involved in the DNA damage response [65]. Besides, while AKT1 activation and phosphorylation are regulated by PIP_3_ at the plasma membrane, the PI3,4P_2_ pool drives AKT2 phosphorylation and GSK3 activation at the plasma membrane and early endosomes [66], suggesting that CdtB-mediated dephosphorylation of PIP_3_ to PI3,4P_2_ results in compensatory perturbations of the AKT/GSK3 pathway. Therefore, in response to CDT intoxination, the p-AKT1 level could be simultaneously decreased by PIP_3_ dephosphorylation and increased in response to DNA damage, whereas p-AKT2 could increase in response to PI3,4P_2_ production. Thus, global assessment of the p-AKT level might be misleading to estimate CdtB phosphatase activity or must be more carefully investigated regarding spatiotemporal regulation and the different AKT isoforms.

Lymphoblastoid cell lines are among the most sensitive to CDT, and their degree of susceptibility has been shown to correlate with PIP_3_ levels [67]. Jurkat cells exhibit strong sensitivity to CDT while containing a high PIP_3_ level due to PTEN and SHIP1 deficiency, two PIP_3_ phosphatases. If CDT susceptibility in Jurkat cells directly depends on CdtB-mediated PIP_3_ dephosphorylation, one would expect that mutations affecting phosphatase activity would greatly reduce CDT cytotoxic effects. Our data indicate that A163R and G114T mutants, supposed to be phosphatase defective, achieve CDT-induced cell cycle defects with the same magnitude in Jurkat compared to HeLa cells, which present a much lower basal PIP_3_ level. Therefore, the higher susceptibility of Jurkat cells to CDT might not depend on CdtB phosphatase activity. However, Jurkat sensitivity could still be explained by the absence of PTEN, as PTEN loss has been shown to inhibit homologous recombination repair and thus increase susceptibility to a genotoxic agent such as mitomycin C [68]. More generally, the numerous tumor-suppressive functions of PTEN in the nucleus that are related to the DNA damage response, including regulation of apoptosis, cell cycle, or senescence, are largely lipid-phosphatase-independent [69]. Thus, the relationship between PTEN status and CDT susceptibility might not only be related to their action in the regulation of the membrane lipid pool but could also depend on PTEN’s role in the CDT-induced DNA damage.

CDT cytotoxic and cyclomodulin activities are tightly linked to DNA damage induced by CdtB nuclease activity [70]. However, the second catalytic activity consisting of PIP_3_ dephosphorylation is also thought to be important for CDT pathogenicity, considering the various signaling functions of this lipid. Although we cannot exclude unanticipated functions during cell intoxination, our results imply that CdtB phosphatase activity is primarily involved in CdtB entry and/or trafficking inside host cells and depends on A163 and G114 residues. CdtB is then addressed to the nucleus through a mechanism independent of its catalytic activities but requiring F146 and T158 in addition to the already identified NLS. Nuclear CdtB then induces DNA damage in an RRN-dependent manner, resulting in cell cycle arrest and eventually, cell death. To conclude, this study supports that the two HducCdtB activities work closely together during cell infection: Phosphatase activity regulates CdtB intracellular delivery by hijacking endocytic trafficking, while nuclease activity is responsible for CDT toxicity by inducing DNA damage. Thus, targeting the phosphatase activity could disturb intracellular trafficking by impacting the PIP_3_ and PI3,4P_2_ concentration. Our data highlight and reconcile the importance of the CdtB dual functionality. Further studies will be necessary to better characterize the spatiotemporal modulation of phosphatidylinositol pools in response to CDT to define more precisely the endocytic routes followed by CdtB.

## 4. Materials and Methods

### 4.1. Bacterial Strains, Cell Culture, Reagents

Cloning was performed in *E. coli* DH5α, protein production performed in *E. coli* BL21 (DE3) (New England Biolabs, Ipswich, MA, USA). Bacteria were grown in an LB-Miller medium supplemented with 30 µg/mL kanamycin (Sigma, St. Louis, MO, USA). HeLa-S3 cells (ATCC) were cultured in Dulbecco’s Modified Eagle Medium (DMEM, Gibco, Gaithersburg, MD, USA) and Jurkat E6-1 (ATCC) in Roswell Park Memorial Institute medium (RPMI, Gibco). Both media were supplemented with 10% heat-inactivated fetal bovine serum (FBS, Gibco) and 1% penicillin/streptomycin (Gibco), and cells were cultured at 37 °C in a 5% CO_2_ humidified atmosphere and subcultured every 2–3 days.

### 4.2. Structure and Sequence Analysis

Protein structures from HducCdtB [24], Bovine DNase I [53], and Type II Inositol 1,4,5-trisphosphate 5-Phosphatase [52] (respective Protein Data Bank accession number: 1SR4, 2DNJ, 4CML) were used in order to identify the potent interaction of CdtB amino acid residues with phosphatidylinositol or DNA. Superposition of the protein structures (1SR4 chain B, 2DNJ chain A and 4CML chain A) was realized using the root mean square deviation (RMSD) tool of the VMD 1.9.4a42 software program. The 3D-spatial alignment was performed based on the highly conserved catalytic residue asparagine (N33 in 1SR4 corresponding to N7 in 2DNJ and to N273 in 4CML).

### 4.3. Plasmids, Cloning, and Sequence Analysis

HducCdtB mutations were generated with a Q5 Site-Directed Mutagenesis Kit (New England Biolabs) using the pRSF-DUET1 HducCdtB WT plasmid as a template [34] and the primers (Sigma) listed in Appendix A. All HducCdtB sequences were subcloned in the mammalian expression vector pmCherry-C1, in frame with mCherry fluorescent protein, using XhoI and EcoRI restriction enzymes (New England Biolabs). The HducCdtB sequence was amplified with Q5 High-Fidelity DNA polymerase (New England Biolabs) and pmCherry subcloning primers presented in Appendix A. PCR products were purified with the GFX PCR DNA and gel band kit (Illustra-GE Healthcare, Chicago, IL, USA), and ligation was performed with Instant Sticky-end Ligase Master Mix (New England Biolabs). For several pmCherry-C1 HducCdtB constructions, the chromatibody (Cb) sequence was cloned in frame with the CdtB-mCherry. The chromatibody came from pmCherry-C1 chromatibody-RNF8 [57], and the cloning was performed with EcoRI and AgeI restriction enzymes (New England Biolabs). All plasmids were purified with EZ-10 Spin Column plasmid DNA Minipreps Kit (BioBasic, Markham, ON, Canada), and all constructs were analyzed by DNA sequencing (Eurofins Scientific, Luxembourg).

### 4.4. CDT Subunits Purification and Holotoxin Reassembly

CdtA, CdtB, and CdtC subunits were purified under denaturing conditions as previously described [34]. Briefly, overnight bacterial culture was diluted in LB-Miller medium with kanamycin, and protein expression was induced with 100 µM Isopropyl-β-d-thiogalactoside for 3 h at 37 °C, 250 rpm. Cells were collected by centrifugation and lysed both enzymatically, by lysozyme, and mechanically using a Dounce homogenizer (10 strokes) followed by 10 sonication cycles for 30 s separated by 1 min cooldown on ice. Insoluble proteins were collected by ultracentrifugation and washed by detergent buffer (Tris-HCl 20 mM at pH 7.5, NaCl 180 mM, sodium deoxycholate 25 mM, EDTA 2 mM, IGEPAL CA-630 1%) before resuspension in urea buffer (phosphate buffer 20 mM at pH 7.2, urea 7 M at pH 7.5, NaCl 500 mM, imidazole 10 mM). The protein solution was then incubated with Cobalt beads (HisPur Cobalt Resin, ThermoScientific, Waltham, MA, USA), the beads were washed, and bound proteins were recovered by elution with imidazole (phosphate buffer 20 mM at pH 7.2, urea 7 M at pH 7.5, NaCl 500 mM, imidazole 60 mM). Denaturated proteins were stored at −80 °C.

CDTs were reconstituted by co-refolding of the 3 subunits mixed at an equimolar concentration in 8 kDa cut-off dialysis cassettes (GeBAflex, Gene Bio-Application, Yavne, Israel). Refolding of CdtB subunits or CDT holotoxin was performed by successive dialysis to gradually remove urea (dialysis protocol adapted from [71]): 12 h in each buffer with decreasing urea concentrations (HEPES 20 mM at pH 7.5, NaCl 200 mM, MgCl_2_ 4 mM, CaCl_2_ 4 mM, glycerol 5%, L-arginine 400 mM, urea 4 M or 2 M or 0 M) followed by a last 12 h dialysis in conservation buffer (HEPES 20 mM at pH 7.5, glycerol 5%). Finally, proteins were flash-frozen and stored at −80 °C.

Holotoxins and subunits samples were boiled in 1× Laemmli, separated by SDS-Polyacrylamide Gel Electrophoresis (SDS-PAGE), and quantity and purity were determined by Stain Free Technology (TGX FastCast Acrylamide gel, BioRad, Hercules, CA, USA) with the ChemiDoc Imaging System (BioRad) using bovine β-casein as a standard.

### 4.5. Clonogenic Assay

Clonogenic assays were performed as previously described [20]. HeLa cells were plated in triplicate at a density of 300 or 3000 cells per well in 6-wells plate. Cells were treated with holotoxins at the indicated concentration 1 day after plating and grown for 8 to 12 days. Formed colonies were fixed for 10 min in methanol 10% (*v/v*)-acetic acid 10% (*v/v*) and then stained for 10 min with 1% crystal violet (*wt/v*) in methanol. Plates were washed by immersion in water and dried at room temperature. Colonies with more than 50 cells were counted, and the surviving rate was calculated in comparison with the non-treated condition.

### 4.6. Cell Transfection

Cell transfection was performed as previously described [34]. Briefly, 40,000 HeLa cells were seeded in 24-wells plates with or without glass coverslips for immunofluorescence or flow cytometry, respectively. Plasmid transfection was performed 1 day after plating with TransIT-2020 (MiriusBio, Madison, WI, USA) for 14 h, according to the manufacturer’s instructions. For protein transfection, CdtBs were transfected one day after plating with TransIT-X2 (MiriusBio), according to the manufacturer’s instructions (concentration and incubation time are specified in corresponding figures).

### 4.7. Immunofluorescence

HeLa cells were fixed in 4% paraformaldehyde for 20 min, washed twice in Phosphate Buffered Saline (PBS) and permeabilized with PBS Triton X-100 0.5% for 10 min. After 2 washes in washing buffer (PBS, NP40 0,1%), cells were blocked for 1 h in washing buffer supplemented with 3% Bovine Serum Albumin (BSA). Cells were incubated with mouse anti-γH2AX antibody (Epitomics, Burlingame, CA, USA) diluted 1/1000 in blocking buffer for 1 h, before 3 washes in washing buffer. Cells were then incubated for 45 min with Alexa 488-conjugated goat anti-mouse secondary antibody diluted 1/800 in washing buffer. Cells were washed 3 more times in washing buffer, and nuclei were counterstained with 4′,6-diamidino-2-phénylindole (DAPI) for 10 min before mounting in p-Phenylenediamine (PDA). Images were acquired with a fluorescent microscope (Eclipse 50i, Nikon, Tokyo, Japan) and were analyzed using ImageJ software.

For each holotoxin and protein transfection experiment, 100 to 300 cells per condition were imaged. The γH2AX signal intensity of each nucleus was automatically determined by an ImageJ macro. First, the γH2AX signal intensity of the whole cell population was averaged for each condition, and these results were normalized to 1 for the untreated condition. In parallel, images used for the untreated condition were manually analyzed to determine which cells presented more than 10 γH2AX foci. Based on these cells, a positivity threshold was determined for the whole experiment and automatically applied to the different conditions to calculate the proportion of γH2AX-positive-cells.

For each plasmid transfection experiments, 30 to 100 cells were imaged, the nuclear and cytoplasmic mCherry signals were manually determined with ImageJ, and the 3 experiments were pooled together for statistical analyzes.

### 4.8. Flow Cytometry

HeLa cells were trypsinized, resuspended in DMEM-SVF 10%, and washed in PBS while Jurkat cells were harvested by centrifugation. Both cell lines were fixed in 4% paraformaldehyde for 20 min, then washed in PBS. After permeabilization in PBS-Triton X-100 0.5% for 10 min, cells were counterstained by DAPI. Cell cycle profiles were acquired with Miltenyi MACSQuant Analyzer 10 cytometer, and data were analyzed with FlowLogic software (Inivai Technologies, Mentone, Australia).

### 4.9. Western Blots

Jurkat cells were harvested by centrifugation, washed in PBS, and resuspended in 1,5X Laemmli. Cells were then sonicated for 10 s and boiled at 90 °C for 5 min. Proteins were separated on 10% acrylamide SDS-PAGE and transferred to a nitrocellulose membrane (Amersham, Little Chalfont, UK). Membranes were blocked for 1 h with blocking buffer (Tris-HCl 20 mM pH 7, NaCl 150 mM (TBS 1×) 50/50 (*v/v*) with Odyssey Blocking Buffer (Rockland, Rockland, MA, USA), Tween 20 0.05%, sodium azide 0.02%) and then incubated with primary antibody diluted in blocking buffer. AKT was detected by 1/1000 rabbit anti-AKT antibody (Cell Signaling, Danvers, MA, USA), p-AKT by 1/1000 mouse anti-p-AKT S473 antibody (Cell Signaling) and GAPDH by 1/5000 rabbit anti-GAPDH antibody (GeneTex, Irvine, CA, USA). After 3 washes in TBS 1×, Tween 20 0.5%, membranes were incubated with secondary antibody diluted in washing buffer: 1/50,000 peroxidase-conjugated AffiniPure Donkey antibody (Jackson ImmunoResearch, West Grove, PA, USA) or 1/5000 fluorescent goat antibody CF770 (Biotium, Fremont, CA, USA). Membranes incubated with peroxidase-conjugated antibodies were revealed by Clarity Western ECL Substrate (BioRad) with the ChemiDoc Imaging System (BioRad) and membranes incubated with fluorescent antibodies were imaged with an Odyssey Infrared Imaging Scanner (Li-Cor ScienceTec, Lincoln, NE, USA).

### 4.10. Data Analysis

The results were expressed as the mean ± SD of at least 3 independent experiments. Statistical analyses were performed with the Prism 8 software (GraphPad Software Inc., San Diego, CA, USA). One-way ANOVA followed by Dunnett multiple comparison tests were used. Differences were considered significant at *p*-value < 0.05 and were indicated by asterisks for comparisons with the untreated condition, hashtag for comparisons with the WT condition and dollar sign for comparisons between Chromatibody/non-Chromatibody conditions (pmCherry transfections experiments): *, # or $ for 0.01 < *p* < 0.05; **, ## or $$ for 0.001 < *p* < 0.01; ***, ### or $$$ for 0.0001 < *p* < 0.001; ****, #### or $$$$ for *p* < 0.0001.

## Figures and Tables

**Figure 1 toxins-12-00530-f001:**
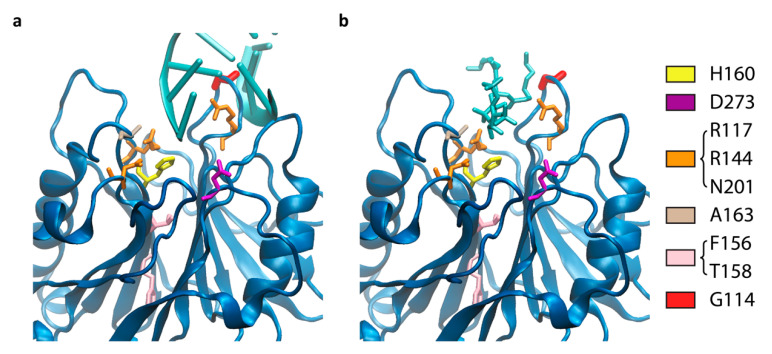
Position of the mutated residues on HducCdtB structure regarding DNA or lipid ligands. The HducCdtB catalytic domain is represented with two putative ligands, octameric DNA (**a**) or phosphatidylinositol-3,4-bisphosphate (**b**). H160: Yellow; D273: Purple; R117, R144, and N201: Orange; A163: Ochre; F156 and T158: Pink; G114: Red.

**Figure 2 toxins-12-00530-f002:**
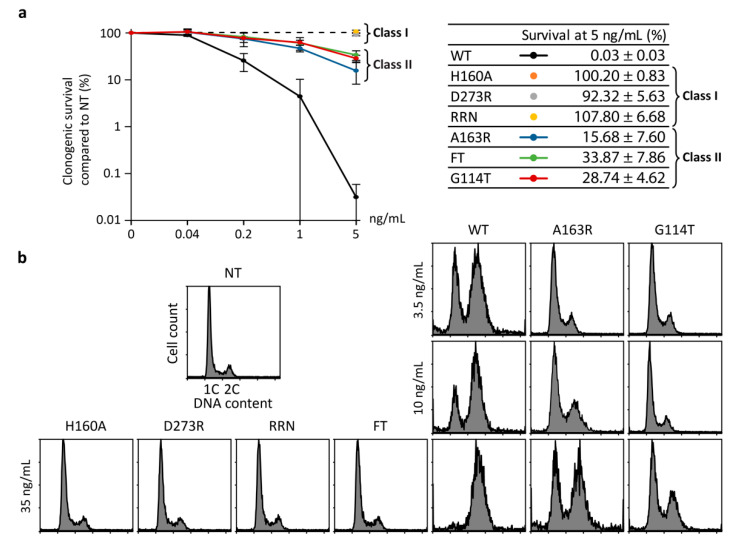
Effect of mutations on CDT-induced HeLa cytotoxicity. (**a**) Clonogenic survival of HeLa cells after 10 days of treatment with HducCDT mutants at the indicated concentrations. The table presents the mean survival at 5 ng/µL ± SD of three independent experiments; (**b**) representative cell cycle analysis of HeLa cells after 36 h exposure to HducCDT WT and mutants at the indicated concentrations. NT: Untreated cells.

**Figure 3 toxins-12-00530-f003:**
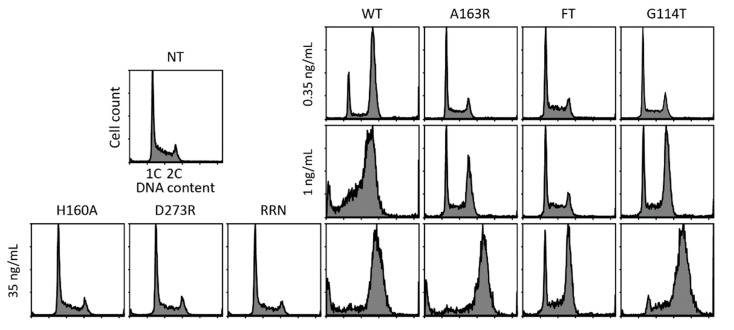
Effect of mutations on CDT-induced Jurkat cytotoxicity. Representative cell cycle analysis of Jurkat cells after 36 h exposure to HducCDT WT and mutants at the indicated concentrations. NT: Untreated cells.

**Figure 4 toxins-12-00530-f004:**
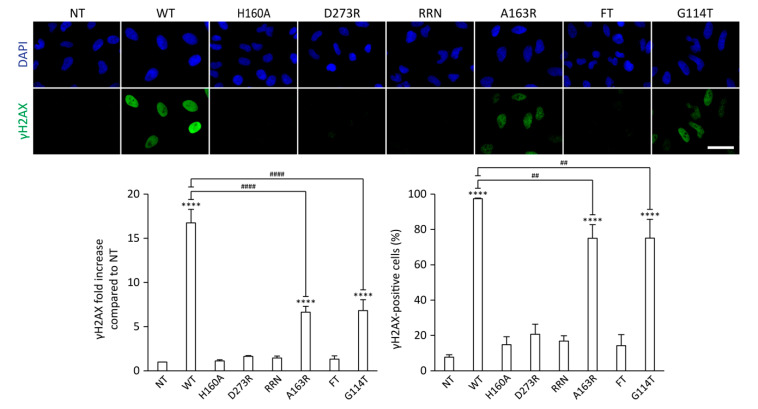
Effect of mutations on CDT-induced DNA damage. Representative images of γH2AX immunofluorescence and associated γH2AX quantifications from HeLa cells treated with 35 ng/mL of HducCDT mutants for 24 h. Quantifications of γH2AX signal are represented as the mean fluorescence intensity per cell (normalized to 1 for the untreated condition) or as the proportion of γH2AX-positive cells. Results present the mean ± SD of at least three independent experiments. Statistical differences were analyzed between each CDT treatment and the untreated condition (****, *p*-value < 0.0001) and between CDT mutants and the WT CDT (##, 0.001< *p*-value < 0.01; ####, *p*-value < 0.0001). NT: Untreated cells. Scale bar: 50 µm.

**Figure 5 toxins-12-00530-f005:**
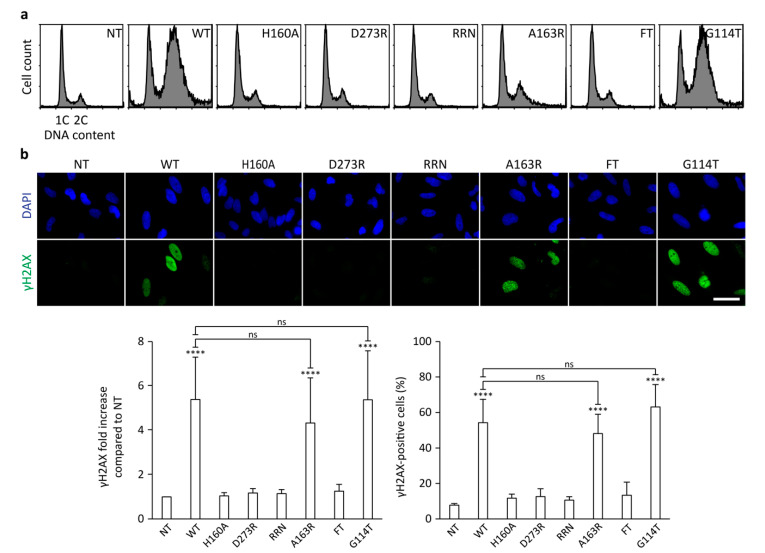
Effect of mutations on CdtB-induced cytotoxicity and DNA damage. (**a**) Representative cell cycle analysis of HeLa cells transfected with 1 ng/mL of HducCdtB mutants for 36 h. (**b**) Representative images of γH2AX immunofluorescence and associated γH2AX quantifications from HeLa cells transfected with 1 µg/mL of HducCdtB mutants for 24 h. Quantifications of γH2AX signal are represented as the mean fluorescence intensity per cell (normalized to 1 for the untreated condition) or as the proportion of γH2AX-positive cells. Results present the mean ± SD of at least three independent experiments. Statistical differences were analyzed between each CDT treatment and the untreated condition (****, *p*-value < 0.0001) and between CDT mutants and the WT CDT (ns, *p*-value > 0.05). NT: Untreated cells. Scale bar: 50 µm.

**Figure 6 toxins-12-00530-f006:**
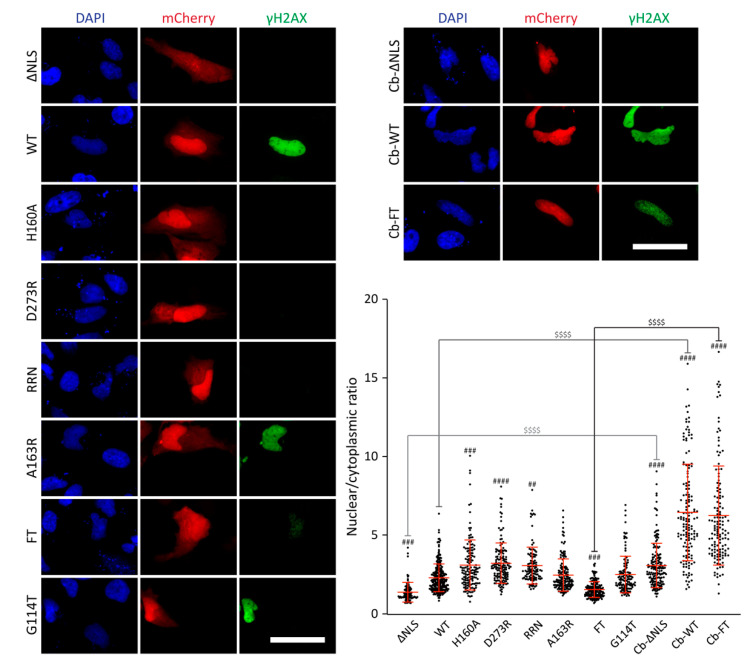
Correlation between CdtB intracellular localization and DNA damage induction in cells expressing CdtB. Representative images of γH2AX immunofluorescence, mCherry fluorescence, and associated mCherry signal quantifications from HeLa cells transfected with pmCherry-C1-HducCdtB mutants with or without chromatibody (Cb) for 14 h. Quantifications are represented as the ratio for each individual cell of the nuclear over cytoplasmic mCherry signal. Results present every cell analyzed over three independent experiments and the mean ± SD of these cells. Statistical differences were analyzed between CDT mutants and the WT CDT (##, 0.001 < *p*-value < 0.01; ###, 0.0001 < *p*-value < 0.001; ####, *p*-value < 0.0001) and between the chromatibody-fused mutants and the unfused corresponding mutants ($$$$, *p*-value < 0.0001). NT: Untreated cells. Scale bar: 50 µm.

**Table 1 toxins-12-00530-t001:** List of HducCdtB mutants used in this study. The putative function of targeted residues and associated references are noted.

Mutation	Function	Reference
H160A	Catalytic	[40]
D273R	Metal binding	[54]
R117A-R144A-N201A (RRN)	DNA binding	[24]
A163R	Putative inositol binding	[49]
F156I-T158I (FT)	Putative phosphatase activity	[49]
∆114–124 (∆NLS)	Nuclear Localization Signal	[38]
G114T	Putative lipid binding	This publication

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
