# Peer review of "Functional Study of *Haemophilus ducreyi* Cytolethal Distending Toxin Subunit B"

_toxins, 2020, doi:10.3390/toxins12090530_

Round 1

Reviewer 1 Report

Functional Study of Haemophilus ducreyi Cytolethal Distending Toxin subunit B.

In this manuscript the authors seek to generate separation of function mutants to understand the importance of proposed nuclease and phosphatase activities of CDtB in cells.  To this end, they utilize structural comparisons to DNAse I and INPP5B in conjunction with past literature to identify residues that can be mutated to achieve this goal.  These residues lead to the identification of two classes of mutation by their impact on cellular proliferation.  Interestingly, these classes also segregate by the function they are proposed to perturb.  Surprisingly, while one mutant, FT, fell into class II with other putative phosphatase mutants in terms of viability, it does not behave like them in terms of cell cycle and DNA damage impacts.  This prompted the authors to explore potential effects on localization as a factor in the divergent effects of these mutants.  The findings of these studies are intriguing and the authors include thoughtful discussions of their implications.  However, there are a few areas where conclusions should be revisited or where additional data are needed to substantiate the interprations.   

Major Concerns:

  1. Somewhat surprising in Fig 2 that the FT mutant behaves like class I mutants in terms of cell cycle, but impacts survival like class II mutants, whereas in 2 it behaves more like class II mutants in the cell cycle of Jurkat cells. The authors should determine the impact of treating Jurkat cells with FT on cell growth. While later assays explain why the FT mutant behaves differently than the other class II mutants in terms of cell cycle arrest and DNA damage induction, these data do not explain why this mutant is as toxic as the class II mutants. The authors should acknowledge and discuss point. 
  2. The authors should treat cells with higher concentrations of the mutants in growth assays to support their assertion that class II mutants are 10-fold less potent than the WT. While likely true, there is not data demonstrating that the class II mutants decrease cytotoxicity in jurkat cells. The authors should reword the statement on line 178 to reflect this. 
  3. It is not clear what constitutes a γH2AX-positive cell in Fig 3. As many cancer cells, including Hela, have a background of foci, this distinction is confusing.  γH2AX is thus analyzed as a number of foci per nucleus rather than positive/negative.  Similarly, it is not clear how the intensity of nuclei can be set to 1 for control cells.  If damage is measured per nucleus, there should be a range of values for each population. This could be an important point as it may suggest that there elevated damage levels in the FT mutant that might be expected given the partial mislocalization of this protein.
  4. There appears to be a disconnect between 10x reduction in cytotoxicity for class II mutants and impacts on DNA damage. Perhaps this is due to using 35 ng/mL in these assays vs the much lower concentrations used for viability.  This point should be addressed.
  5. The assertions of trafficking defects described in section 2.4 need substantiation. The authors should show intracellular levels of CdtB mutants achieved by holoenzyme delivery and transfection.  This data would also strengthen the interpretation of data in Fig 5.
  6. The authors state on line 262 that directing the FT mutant to the nucleus fully restores the induction of DNA damage. This statement is based on a single image of one transfected cell, which is not sufficient.  To support this statement, quantitation such performed in Figs 4 and 5.  The data would also seem to suggest that it is the ability to bind DNA, not just nuclear localization, that leads to DNA damage.  As one would expect that there would be some level of DNA damage induced by this mutant given that it is partially nuclear, at least as judged by the single example shown.  It does not appear to be nuclear excluded.  It would be interesting to test this further by determining whether fusing a heterologous NLS to the FT mutant would not impact induction of DNA damage.  This idea could then be further tested by determining whether the CB-RRN mutant is able to induce damage and similarly that forcing WT to the cytoplasm would rescue the effects.  Such experiments are necessary to substantiate statements such as that made on line 318 that the inability to induce DNA damage is due to reduced nuclear localization.  Nuclear targeting and chromatin targeting (as mediated by the chromatibody) are not necessarily the same thing. 

Minor Concerns:

  1. The authors might consider adding identifying labels or even a simple legend into the Fig 1. The same applies to Fig S1.
  2. The concentrations of HducCDT in Fig. S3 b and c are not indicated as suggested in the figure legend.
  3. The description of panel a in Fig S4 is not accurate.

Reviewer 2 Report

In overall its quite an interesting study, however, it needs some improvements and I have some comments:

Introduction:

Lanes 37-39: It’s not entirely clear which subunit (or subunits) of CDT (CdtA, CdtB or CdtC) form A and B2 parts of the AB2 complex of the holotoxin.

Results:

Lanes 131-132: "The purified WT and mutants CdtBs (Fig S2) were incubated with CdtA and CdtC subunits to reassemble active holotoxins". I missed the description in the Methods section, how CdtA and CdtC proteins were produced. Did authors apply some methods to determine if all three subunits were assembled into holotoxin successfully?

Fig. 2a : What is indicated on the X-axis of the graph?

Lane 184: "To further compare the defects associated to CdtB mutations, we evaluated their impact on DNA 184 damage induction by monitoring γH2AX increase, <...>." It would be nice if authors would provide a reference (here or in the methods), were monitoring of γH2AX is described as a method to determine DNA damage.

Fig. 4 : The abbreviation "NT" is not explained in the caption of the figure. 

Discussion:

Lanes 288-289: "Here, we propose to use cell-based approaches to estimate the impact of CdtB mutations on host cell responses and relate them to CdtB catalytic activities." Well, cell-based methods were already proposed by Pons et al., 2019 (reference 34 in the manuscript). And much of this work was conducted based on methods described in this reference. 

Materials and Methods:

All this section lacks references.

Lane 427: Could authors specify about dounce-sonication cycles used for cell disruption?

Lanes 429: What is the composition of detergent and urea buffers mentioned in the text?

Lane 444: Which cells were plated? E. coli? HeLa, or Jurkat?

Reviewer 3 Report

An overall objective of this study entitled Functional study of Haemophilus
ducreyi Cytolethal Distending Toxin subunit B
was to characterize the role and interplay between CdtB nuclease and phosphatase activities.

There are minor points need to be clarified

Quantification of immunostaining is considered as semi-quantitative. Here, most of the results from immunofluorescence correspond well to cell cycle analysis data therefore additional testing is not necessary. However in the case of Figure 5, there is the difference between the data from cell cycle analysis of A163R and A163R immunofluorescence staining, therefore more analysis would be helpful (e.g. expression of gH2AX analyzed by western blot). 

For statistical analysis, standard deviations should be used instead of standard errors.

Do you have any explanation why you did not observe any alteration of p-AKT after CDT in Jurkat cells?

Figure 1: the color of mutated residues is not clear and bright, ochre is almost invisible, orange looks similar to ochre, pink is very light, and red looks like the orange color.

Figure S1: Explanation of the left and right panel is missing.

Figure 2: a) Labeling of axis x is missing. 

Figure S3: b,c) What is indicated concentration of HducCDT mutants and wt? e) The explanation of AT is missing.

Figure 5b: There should be chosen images with similar DAPI FI.

Figure 6: Quantification of gH2AX signal should be presented, especially the proportion of gH2AX-positive cells.

Round 2

Reviewer 1 Report

The authors have added additional data and methods details and have modified the text to sufficiently address concerns from the previous version.